# Use of One-Stage Detector and Feature Detector in Infrared Video on Transport Infrastructure and Tunnels

David Švorc [1], Tomáš Tichý [2], Miroslav Růžička [1,*] and Petr Ivasienko [2]

[1] Faculty of Engineering, Czech University of Life Sciences Prague, 165 00 Prague, Czech Republic
[2] Faculty of Transportation Sciences, Czech Technical University, 110 00 Prague, Czech Republic
* Correspondence: ruzicka@tf.czu.cz

**Abstract:** This article presents the use of the combination of the object detection method and feature detector in an infrared video on traffic infrastructure and in tunnels. The theme of the paper is the validation of vehicle detection and its classification using infrared video streams. In addition, the article focuses on the use of a feature detector and object detection to distinguish between vehicles with electric and combustion motors. The method suggests the use of a low-resolution thermal camera as an inexpensive extension of installed thermal camera technologies. The developed system has been verified for the applicability of vehicle detection and classification using object detection methods and their application in transport infrastructure and tunnels. It also presents a method for distinguishing propulsion units into electric and internal combustion; both systems' conclusions are then statistically verified. The application of the system is evident in regional traffic management systems, including safety applications for traffic control in tunnels. Categorizing vehicles provides valuable information for higher levels of traffic management, toll systems, and municipal database systems, as well as for a preventive system for estimating vehicle conditions and their potential of fire in tunnels.

**Keywords:** thermal detection; diagnostics methods; traffic control and ITS; tunnel systems; vehicle classification; CNN; electric vehicles

## 1. Introduction

Many techniques for detecting, tracking and categorizing objects in images have been developed in recent years to optimize reliable vehicle detection and classification [1–4]. Modern traffic management and ITS (Intelligent Transportation System) use these advanced functions. Due to changeable weather and light conditions, these applications require uninterrupted functionality [5]. The widely used visual cameras still have low accuracy in bad weather conditions e.g., fog, snow, or heavy rain [6–9]. Consequently, visual cameras with infrared cameras can replace or supplement conventional methods [7]. Due to the physical nature of infrared thermography, infrared cameras can offer continuous detection of vehicles regardless of the surrounding conditions. Combining thermal video cameras with image recognition techniques provides an interesting alternative to other sensors [10].

Conventional detectors placed on infrastructure represent other ways of detecting vehicles through the use of the principle of detection loops, ultrasound, and radar principles or most often video detection. Cooperative systems (C-ITS) represent a potential for a suitable and reliable detection. These systems use vehicle-to-vehicle communication or appropriate communication with the infrastructure. The infrastructure manager receives accurate information about the vehicle (the vehicle status), and he can send the required information, navigation, and control data or other information [11].

These approaches and solutions are applied on conventional transport infrastructure, in road tunnels, especially by more rugged tunnels in the case of longitudinal gradients. The use of thermal imaging detection, which is introduced below, has its significant safety

justification by longitudinal gradients. Although tunnels are equipped with detection systems based on video detection or special line cable technology connected to the fire detection system [12], detection using thermal imaging in the tunnel can reveal potential causes of fire based on overheating of some parts on the vehicle (wheel discs, cargo, etc.). An early detection and the use of thermal imaging in tunnel system algorithmization have a fundamental potential for higher safety.

Detecting features and characters in an image is a fundamental step in many computer vision applications which provide abstractions of image information with respect to only selected salient features. A good feature detector will provide features that are salient, invariant, and resolving [13]. A key problem of computer vision is that object detection belongs to the basics of other computer vision tasks, such as image segmentation [14], image capture [15], and object tracking [16]. From an application point of view, object detection offers two research topics: "general object detection" and "applications of the detection". General object detection aims at exploring methods for detecting different types of objects in a unified framework for simulating human vision and recognition. Applications of the detection focus on specific application scenarios, such as vehicle detection, pedestrians (objects in general), text detection, etc.

The article presents the system of detection and categorisation of vehicles into Battery Electric Vehicles (BEVs) and Internal Combustion Engine Vehicles (ICEVs) in the thermal video stream running on a combination of a one-stage detector RetinaNet and the Maximally Stable Extremal Region method (MSER). RetinaNet detects and classifies vehicles and is supplemented by the picture database. The proposed system subsequently used the properties of the thermal cameras, i.e., the ability to display warm objects in a grayscale image; these images are then used by the MSER algorithm, which can detect such areas. The detection by MSER algorithm is based on an assumption that the existing heated parts of the ICEVs can be adequately displayed on a thermal image. Subsequently, the difference in the experiments on the number of detected areas of BEVs and ICEVs was statistically evaluated. The results confirm a possible connection of RetinaNet and the MSER algorithm for the detection, categorization, and potential prediction of vehicle states.

The following section will discuss the approaches of object detection using convolutional neural networks and object resolution based on their heat signatures. Section 3 explains the study's methodological framework, research questions, and the tools applied for data analyses including the statistical evaluations. Section 4 presents a comparison of the results and statistical comparison. The results are discussed in the Section 5. Section 6 provides conclusions as to the study findings and their practical implications.

## 2. Related Work

This section deals with the potential of machine learning applications in vehicle detection and heat field detection.

### 2.1. Application of Machine Learning in Vehicle Detection

The use of convolutional neural networks for object detection was considered inefficient for a long time, mainly due to real-time use [17]. Thanks to the development and technological advancement of graphics cards and other computing devices and advanced improvements in the acceleration of the computing power of neural networks, brought in 2012 by Krizhevsky [18] with his "AlexNet" convolutional network, the use of CNN for object detection has been enabled to a greater extent. Figure 1 is a timeline of the development of the most important methods used in object detection.

The use of CNN achieved significant success in vehicle object detection [19]. CNNs have a strong ability to learn image features and can perform various related tasks such as classification and regression of rectangles bounding objects referred to as "bounding box" [14]. The detection method can be divided into two categories. The two-stage method generates a candidate object box using different algorithms and then classifies the object using a convolutional neural network. The one-stage method does not generate a

candidate box, but directly converts the problem of placing the box bounding the object into a regression problem for processing [20]. The two-stage method uses Region-CNN (R-CNN) [21] to selectively search for regions in the image. The image inputted to the convolutional network must have a fixed size, and the deeper network structure requires a long training time and bigger storage memory. SPP NET [22] draws on the idea of spatial pyramid matching and allows images of different sizes to be inserted into the network and have fixed outputs. R-CNN, FPN, and Mask RCNN have improved the feature extraction methods, feature selection, and classification capabilities of convolutional networks in various ways [20].

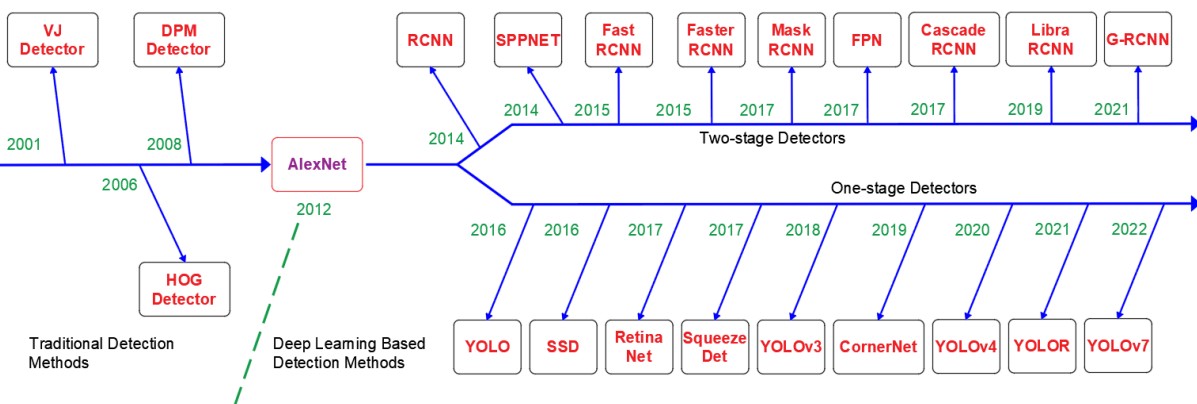

**Figure 1.** The milestones in object detection [17].

The most important among single stage methods are "Single Shot Multibox Detector" (SSD) and "You Only Look Once" (YOLO) [20]. The term 'Single Shot' means that localization and detection are made in a single forward pass of the network at the inference time—the network only has to "look" at the image once to predict [23]. This is also true for the YOLO network, as the full name of this detector implies. YOLO network method achieves processing speeds that enable real-time detection. Its disadvantage is a lower accuracy compared to two-stage detectors.

Another notable single stage detector includes RetinaNet [24]. This method has many similarities with previous detectors such as the RPN (Region Proposal Network) detector [25], SSD [26], and FPN (Feature Pyramid Networks) [27]. In particular, it shares the use of "anchors" with these methods as in RPN, and the use of pyramid features as in SSD and FPN.

RetinaNet is a uniform network named after its dense sampling of object locations in the input image. The design is characterized by an efficient pyramid of features in the network and the use of anchor boxes [25]. It consists of a backbone network and two subnetworks which are responsible for specific tasks. The Focal Loss function is used in a method of one-stage detector RetinaNET [24] to improve the detection results.

The backbone network of Figure 2a is responsible for computing the convolutional map of features over the entire input image. The first subnetwork (see Figure 2c) performs convolutional classification of objects in the output of the backbone network; the second subnetwork (see Figure 2d) performs convolutional regression of bounding boxes.

The reasons why methods like SSD are less accurate than two-stage methods and propose a solution to eliminate this problem are explained in [24]. The improvement is achieved by modifying the loss function. This makes it faster than two-stage methods, though equally accurate and enables real-time object detection applications.

A partially supervised CNN was used to learn good features for vehicle classification in the system proposed by [28]. Softmax regression was used to classify the features obtained by CNN. When comparing their work with other methods, the results show that their method classifies the type of vehicle with at least 92 per cent accuracy. The authors in [29] used a CNN for vehicle detection using image data from low-resolution

cameras. Thus, they demonstrated that, compared to CNN applications in face detection where high-resolution cameras are used, vehicles can also be detected with high accuracy (about 95 per cent) from low-resolution cameras. The system proposed in [30] used "Faster" RCNN for vehicle detection. The use of appropriate neural network tuning parameters and modifying the algorithms significantly improved the vehicle detection results over previous studies.

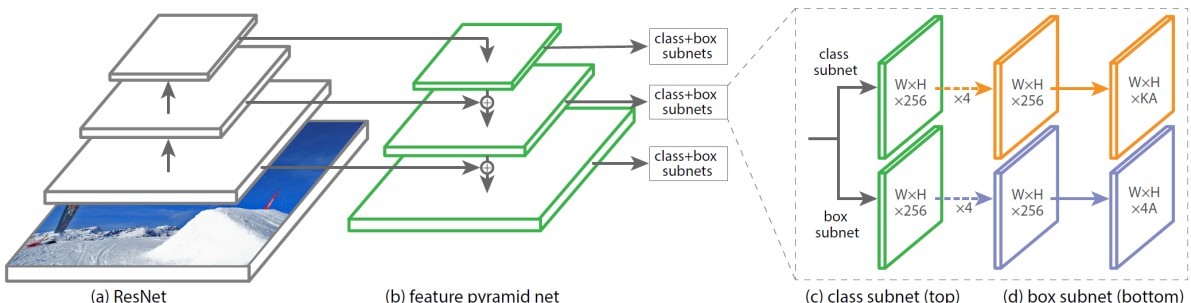

**Figure 2.** RetinaNet network architecture [24].

The researchers in [31] proposed a vehicle detection scheme based on a multi-task deep CNN. The experimental results using a standard test dataset demonstrated a better performance compared to other methods. The authors proposed a depth model for vehicle detection consisting of feature extraction, deformation processing, occlusion processing, and training classifiers using a back-propagation algorithm in the article [32]. A multi-vehicle detection method which consists of a YOLO neural network within a "Darknet" framework was proposed in [33].

### 2.2. Detecting Thermal Fields on an Objects

Only a few authors have addressed the possibility of using thermal properties for object detection [4]. The authors of [2] used visual and thermal cameras for vehicle detection and the area of the radiator grille and headlights was identified as the detection object on vehicles. Other researchers [1,29] used a detection area in the windshield of vehicles and their surroundings. However, this method was not very accurate under winter conditions due to the similar windshield and ambient temperatures [34–36].

Every vehicle generates heat during operation [4]. Even BEVs generate heat during operation, although it is much less than a vehicle equipped with an internal combustion engine. The heat is emitted into the vehicle's body and is easily visible on the front mask and the side of the vehicle. Other parts that do not come into direct contact with the heat from the engine are significantly colder [37]. The article [4] outlines the possibilities of classifying vehicles according to their heat signatures. The authors focused on different thermal characteristics in different vehicle categories and distinguished them by a statistical evaluation of their thermal images. The detection area was the whole image of a predefined height and width. Only a part of the thermal histogram that had been statistically been proven as a representative part of the vehicle category was considered.

The authors in [37,38] stated that the electric motor and the components required for its operation are heated much less compared to the internal combustion engine. This difference might be a basis for detecting BEVs. Vehicle detection is expected in locations such as tunnels where most vehicles will have reached operating temperature. A low number of vehicles with lower than operating temperature will not affect the overall result in the predefined area. However, this is not the only place where there is a significant difference between the temperature field of an ICEV and BEV. The rear of the car at the exhaust outlet are the areas at which to measure other temperature differences because BEVs do not have this system [2,32].

Several scientific papers describe temperature maps of vehicle side profiles [33], detection of small boats on water surface [39], or human detection based on the temperature field

emitted by human bodies [40]. Authors in [41] detected vehicle occupants using thermal and IP cameras using object detection methods. For the detection of boats and people that emit heat, the MSER algorithm was used to detect the maximum stable extremal regions, represented by the bright areas of the image, since most thermal cameras use bright pixel values for the warm regions of the image.

The MSER method was first introduced in [42] to find matches between image features from two images with different viewing angles. MSER has been used in a wide range of computer vision tasks, including Wide Baseline Stereo [42], image retrieval [43], object recognition [44], and tracking and 3D segmentation [45]. The MSER detector has also been implemented on Field Programmable Gate Arrays (FPGAs) [46]. Several extensions to MSER have been proposed to enable colour image processing [47], to extend MSER to handle blurred images [48] better, or to enable text detection [49].

Based on the hierarchical MSER detection structure described in [45], an algorithm using the Min- and Max-tree hierarchy [50] to determine MSER regions was introduced in [43]. Extending the idea presented in [51], the algorithm proposed in [43] can be applied to any component tree that exhibits invariant properties.

The limitations of the thermal camera and its recording must be considered. A working vehicle's engine generates heat and reflects it into its surroundings. The exhaust gases generated by a combustion of fuel by the engine are warmer than the ambient temperature. In addition to these temperature fields, there are other sources which generate heat in the environment. Situations might appear where there is also an infrastructure object near the vehicles in a single image such as people moving around the roadway, and other distractions that are sources of temperature traces. Despite a significant development of thermal cameras in recent years, their resolution is still low compared to visual cameras. As a result of that the image is often distorted. The larger the distance, the more distorted the object are. The most frequent faults in thermal imaging are the following: Camouflage, Reflections, Diverse background, and Video noise/degradation of signals.

## 3. Materials and Methods

The FLIR Thermicam 2390 traffic thermal camera [52] was chosen to record vehicles. The camera uses an FPA detector (Focal Plane Array), which is a matrix detector with a large number of detection plates that simultaneously capture the infrared radiation of the entire visible surface, and an uncooled Vox microbolometer (vanadium oxide) that measures in the long-wave infrared (7–14 μm) [53]. The resolution of this camera is $320 \times 240$ pixels and it records the thermal stream in MPEG-4 and H264 format. The recording rate is 30 fps ("frame per second").

The testing camera was placed at a traffic light-controlled intersection at a height of five meters above the roadway on the traffic light mast arm to verify the function; see Figure 3a. Figure 3b shows the assembly construction of the camera on the mast arm of the traffic lights and its connection to the mounting rack for the power supply and data collection. The thermal camera captures the lanes of the road in front of the traffic light itself.

The wiring of accessories providing camera connectivity, power, and communication is presented in Figure 4. Let us summarize the traffic camera parameters:

- The camera is connected to the POE power supply [54] (Power over Ethernet) using a UTP (Unshielded Twisted Pair) cable.
- The POE is connected to a modem [55] with an Ethernet interface and an embedded data SIM card with unlimited data.
- The data is sent via a GSM (Global System for Mobile Communications) interface to the server—the connection is via a remote desktop using a VPN (Virtual Private Network) protocol.

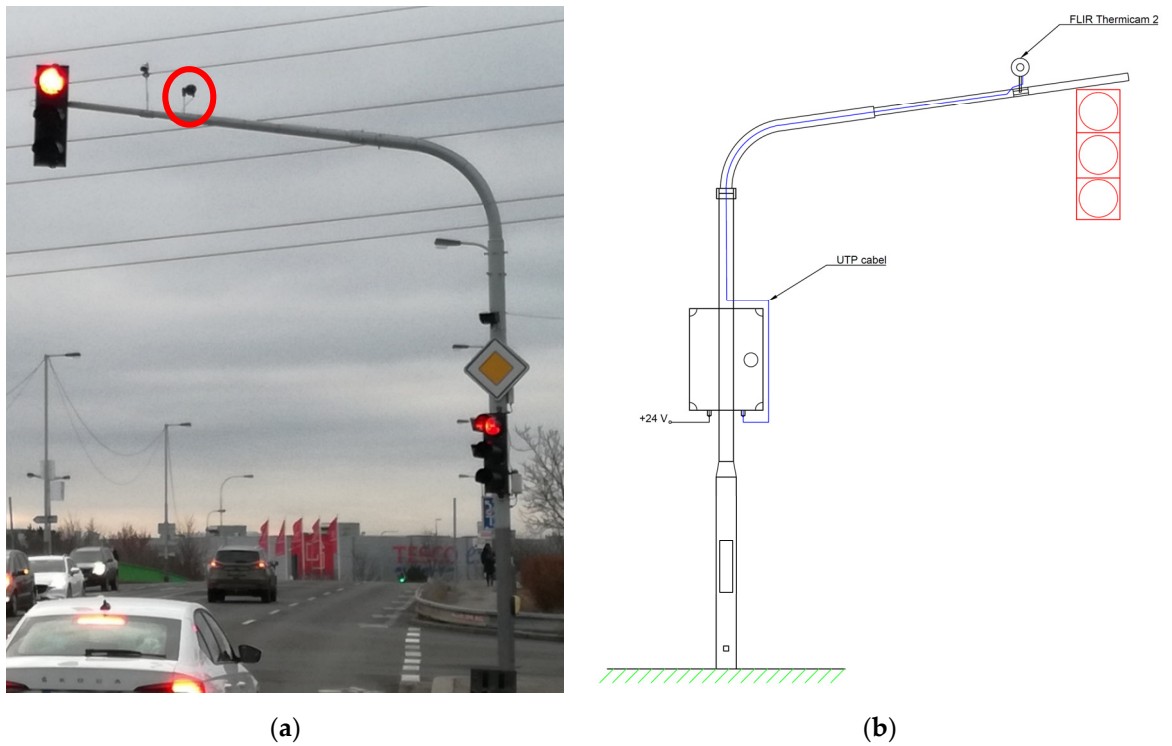

|     |     |
|:---:|:---:|
| (**a**) | (**b**) |

**Figure 3.** The overall installation of a camera and interface on the mast arm of the traffic light; (**a**) installation at the existing traffic light construction, (**b**) scheme of installation.

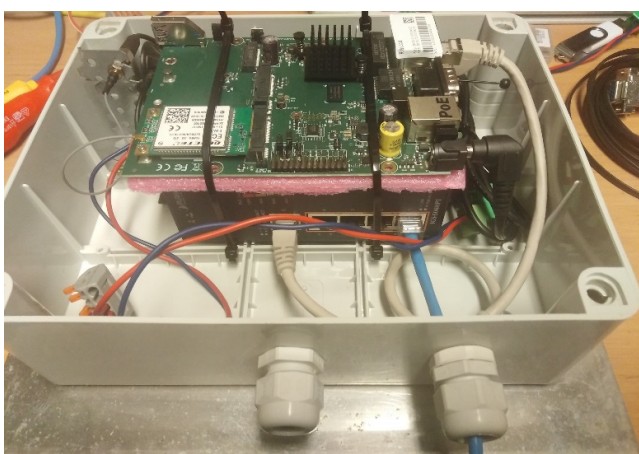

**Figure 4.** Connection of the camera to the power supply including connectivity within the mounting box.

*3.1. Study Hypothesis*

In this paper, the scientific problem was stated as follows: "Is it possible to use the method of object detection for categorization of BEVs and ICEVs based on the thermal fields". The null and alternative hypotheses are namely:

- Hypothesis H0: the ICEVs category does not have a significant difference in the average number of detection windows detected by the MSER algorithm compared to the BEV TESLA.
- Hypothesis H1: The ICEVs vehicle category has a significant difference in the average number of detection windows detected by the MSER algorithm compared to the BEV TESLA vehicle.

### 3.2. Proposed Method of Data Processing

Previous measurements show that the regions of interest in the BEV and ICEV masks are significantly different [37]. Conclusions made in [38] described the possibility of distinguishing between electrically driven BEVs and combustion engine ICEVs. Detection was achieved by targeting vehicle masks, where temperature differences between the two vehicle categories were statistically demonstrated. The differences guided the development of an automatic detection and classification method based on computer vision techniques were presented in [56].

The above-mentioned papers verified the possibility of applying machine learning on vehicle detection and classification from thermal images. Detection was provided using the Haar Cascade algorithm and vehicle classification using CNN which was trained on a database of images acquired by a high-resolution visual camera. Due to the low resolution of the thermal images, it was impossible to train the CNN by the database created from those thermal images. Another issue was long computational time and difficulty in applying real-time detection.

The authors further elaborate and modify the detection and classification method. The low-resolution thermal camera was replaced with the higher image resolution camera introduced at the beginning of Section 3. An object detector RetinaNet was utilized and used to detect vehicles and classify them as ICEVs and BEVs in the infrared video. The above-mentioned detector was selected due to the faster computation time and the possibility of real-time detection. The detector classifies vehicles into following categories: CAR, VAN, and BUS as ICEVs. The TESLA model S was selected as a reference vehicle for the BEV category. The images of vehicles are saved in categories in the database. This method serves as an input to classify the vehicles' propulsions into electric and internal combustion. The maximum stable extremal regions (MSER) detection method was chosen to verify the probability of differentiating vehicles based on temperature fields. The MSER algorithm was used to detect the temperature fields on vehicle masks and to statistically validate its applicability for the differentiation of BEVs and ICEVs.

A thermal video stream was recorded by the traffic thermal camera and sent to the server. It is the input to an object detector that detects and classifies vehicles from the traffic stream and separates different vehicle categories into a database. An image dataset and image annotations were built by the TensorFlow Object Detection API (Application Programming Interface) [57]. The images were annotated by the LabelImg tool [58]. The OpenCV library [59] with computer vision and image manipulation tools were used to improve the created database. The annotated images became training objects in this database. The object detector was trained and used them for classification. Vehicles were classified into four categories.

The ICEVs include categories:

- CAR;
- VAN;
- BUS.

A Tesla Model S was chosen to represent BEVs.

The image dataset was divided into three subsets, namely training, validation, and testing. We defined three sets of dataset splits. The first distribution of the dataset was 90% of the images for training, 3% for validation, and 7% for testing. The second distribution of the dataset was 70% of the images for training, 15% for validation, and 15% for testing. The third distribution of the dataset was 60% of the images for training, 20% for validation, and 20% for testing. These distributions of the dataset were chosen to test the hypothesis of whether it is possible to distinguish between BEVs and ICEVs in a thermal video stream under real traffic conditions using object detection methods. We have taken the liberty to use the first set of dataset distribution because of the expected application of the technology in tunnels or in tunnel complexes. It is expected that the defined distribution of the dataset will meet the requirements of the model and corresponds to its needs for the verification of the above-defined hypothesis. The images used for training, validation, and testing were

captured under outdoor conditions at a light-controlled intersection. Such a location is affected by external meteorological, lighting, and spatial conditions that reduce the quality of the recording. On the other hand, installation in a tunnel for pilot testing and verification is very difficult, as it is necessary to obtain the necessary permits from the authorities.

The pictures in the database became an input for the MSER algorithm. Figure 5 is a scheme to explain the following sections in detail.

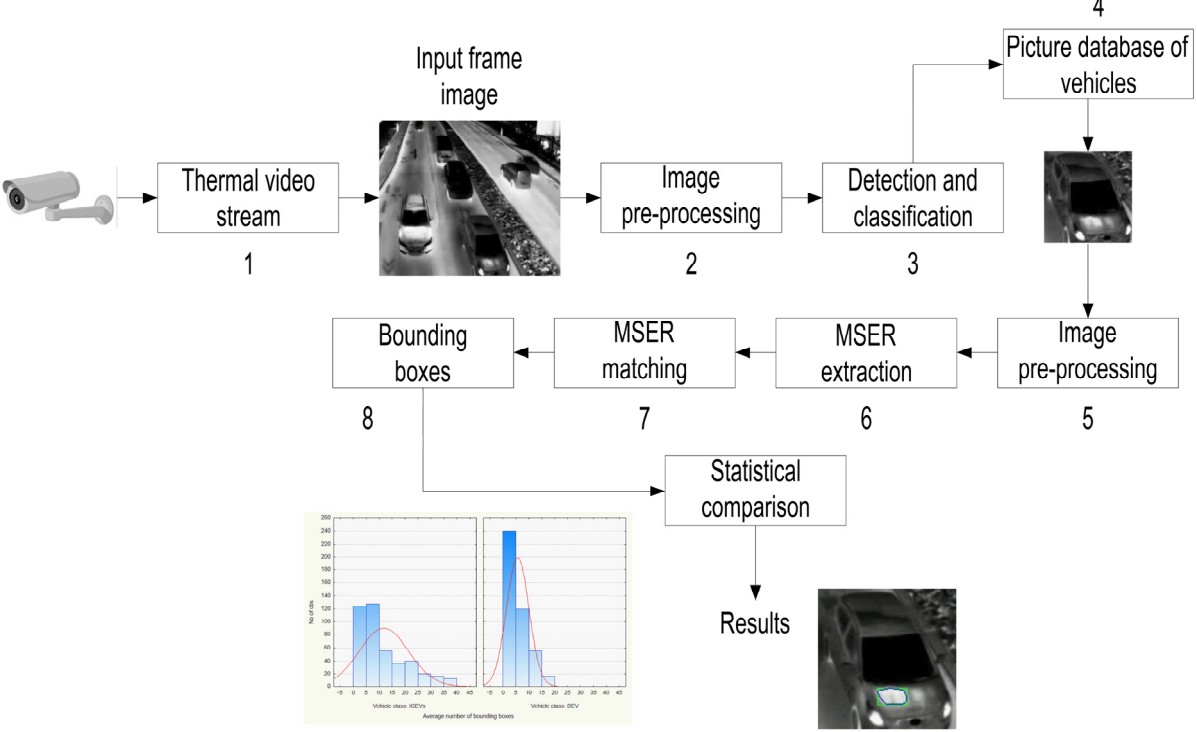

**Figure 5.** Flowchart of proposed method.

1. Assembling the thermal camera and its targeting.
2. Cutting the video stream into individual frames based on FPS recording and pre-processing the frames (resizing) as a modified image input to train the object detector.
3. Object detector training.
4. Creation of vehicle database—breakdown into BEV TESLA and other ICEV categories.
5. The algorithm retrieves an image of a detected vehicle from the database; the image is pre-processed.
6. Detecting MSER regions.
7. Matching MSER regions according to pre-set parameters.
8. Formation of bounding boxes based on convex hulls of MSERs.
9. Statistical comparison of bounding boxes between the BEV and ICEV categories.

### 3.2.1. Object Detection

The method of the one-stage detector RetinaNet was chosen to construct and enable real-time vehicle detection. The detector was created based on the backbone network ResNet [60]. The advantages of this method are mainly high computational speed and high accuracy. The Keras [61] and TensorFlow [57] frameworks were used to build this object detector. In the case of TensorFlow, it is a library that contains procedural tools for building machine learning models, while Keras is an API written in Python.

Object detection was performed using a one-stage object detector, which consists of four parts. Each part performs its task:

1. Backbone network ResNet
   - It is a pre-trained convolutional neural network.
   - It is used to extract features from the input image.
   - It works as a basic feature extractor for object detection.
   - It uses the principle of truncated connections.

2. Model RetinaNet
   - Discretizes the output space of bounding boxes into a set of initial bounding boxes in different aspect ratios and scales according to the location on the feature map.
   - At prediction time, the network generates a score for the presence of each object category in each initial bounding box and adjusts the box to better match the shape of the object.
   - It combines predictions from multiple feature maps with different resolutions to naturally handle objects of different sizes.
   - It uses a loss function, which is a dynamically scaled cross-entropy loss where the scaling factor decreases to zero as confidence in the correct class increases [24].

3. Classification subnet—a subnetwork that extracts object class information from the object detector and solves classification problem.

4. Regression subnet—subnetwork that extracts information about the coordinates of objects in the image from the FPN and solves the regression problem.

### 3.2.2. Image Pre-Processing

We used two methods for image pre-processing before application of the MSER algorithm. The noise removal is applied and image dimensions are adjusted. A detailed description of the methods is as follows:

- Noise removal

To remove noise that may cause false detections and increase the number of detected MSERs, a $7 \times 7$ median filter was used. If the median filter is used, the edges of the images are not blurred since it is not a linear filter.

- Image resizing

The original images are of low resolution, therefore, to use the thermal camera images it will be necessary to increase its original scale; this will slightly prolong the evaluation of the algorithm, but at the cost of better results. The original $320 \times 240$ pixel resolution images were scaled up by a ratio of 1.5.

### 3.2.3. MSER Detection

This method was chosen to validate the functionality of hot field detection because of its computational speed and ease of implementation. The MSER detection method is based on blob detection. It detects features at multiple scales based on image contrast and region intensity.

The idea of blob detection in an image relies on targeting, in computer vision, maximally stable extremal regions. The principle of this method is to target a cluster of pixel information of similar depth and defined density in an image region. This makes it possible to detect image features by identifying not distinct pixels, but entire image structures. This method is applicable on relatively small numbers of regions per image. It is an efficient and fast detection algorithm with a speed close to the frame rate of the video.

MSER algorithm procedure:

1. Maximally stable extremal regions creation;
2. Select MSERs which belong to range of predetermined thresholds parameters;
3. Create bounding boxes;
4. Cull overlapping blobs;
5. Make list of unique regions.

**MSER parameters**

The extremal regions are defined as connected components of pixels that are all of a either higher (maximal region) or lower (minimal region) value than the pixels on the boundary of the region [62]. MSERs are controlled by a single parameter delta (Δ), which controls the behaviour of the stability calculation.

The stability of an extremal region R is the inverse of the relative area variation of the region R when the intensity level is increased by Δ. Formally, the variation is defined as [63]:

$$var = \frac{|R(+\Delta) - R|}{|R|} \tag{1}$$

For our research we used these parameters to detect MSERs:

- Delta: Step size between intensity threshold levels. The more delta increases, the fewer regions are detected.
- Minimal area: Minimum total area of object in pixels.
- Maximum variation: Specify maximum region variation.
- IOU_thresh: IOU (intersection over union) threshold.
- Matrix: Matrix representation of image.
- Ranges of parameters according to Table 1 below were used to create hulls of MSERs.

**Table 1.** MSER's algorithm parameters.

| Parameters | Range of Parameters | Effect of Increase on the Number of Detections |
|:---:|:---:|:---:|
| Δ | 1–10 | decrease |
| Minimal_Area | 400, 500, 600 | decrease |
| Maximum_variation | 0.1, 0.15, 0.2, 0.25 | increase |
| IOU_thresh | 0.7, 0.8, 0.9, 0.95 | increase |

Maximal_area set was based on the matrix size of the picture according to equation:

$$\text{Maximum area} = 0.1 * \pi * \left(\frac{\text{width of the picture}}{2}\right)^2 \tag{2}$$

*3.3. Statistical Metric for Network Evaluation*

The one-stage object detector evaluations was performed by the mAP metric, which is used in applications related to object detection. The calculation of mAP is based on sub-metrics [64]:

- Confusion matrix,
- Intersection over Union (IoU),
- Recall,
- Precision.

It is necessary to first define the four attributes by which the confusion matrix is formed. These are *true positive*, *true negative*, *false positive*, and *false negative*.

The IoU metric indicates the degree of overlap between two bounding boxes. It is therefore used to measure to what extent the predicted area overlaps with the annotated object boundary. The resulting value is compared with the threshold value. This metric can be applied to any algorithm whose output is bounding box predictions. For this metric, it is necessary to know the exact position of the actual bounding box of the object and the position of the predicted box.

The calculation of the IoU value is defined in Equation (3).

$$IoU = \frac{\text{Area of Overlap}}{\text{Area of Union}} \tag{3}$$

The IoU metric is then used to determine true positives. If the proportion is higher than the threshold, the detection is considered successful and annotated object is marked as a *true positive*. Otherwise, it is marked as *false positive*. If the annotated object is not detected, it is marked as *false negative*.

*Precision* indicates the percentage of correct predictions see Equation (4), while *Recall* indicates the correctness of finding all positive results see Equation (5) [65].

$$Precision = \frac{true\ positive}{true\ positive + false\ positive} \tag{4}$$

$$Recall = \frac{true\ positive}{true\ positive + false\ negative} \tag{5}$$

The calculated values of *Precision* and *Recall* are used to construct *precision–recall* curves [65].

Subsequently, the average precision (AP) is calculated. It represents a way to summarize the curve *precision–recall* into a single value representing the average of all precisions.

The general definition of AP is to find the area under the above *precision–recall* curve (see Equation (6)) [66].

$$AP = \int_0^1 p(r)dr \tag{6}$$

Commonly, object detection models are evaluated with different IoU thresholds, where each threshold may provide different predictions from other thresholds. The AP value is used and computed for each class to compute the mAP in a model where there are multiple object categories. The average of the AP for all classes is the mAP value in Formula (7) [65].

$$mAP = \frac{1}{n} \sum_{k=1}^{k=n} AP_k \tag{7}$$

## 4. Analysis and Results

The total number of recorded pictures was 25,690. According to the first defined dataset split, 23,258 images were used to train the network, the verification set contained 586 images, and the test set contained 1846 images. The second distribution of the dataset contained 17,983 pictures to train the network and 3854 pictures for verification and testing the network. The third split resulted in the distribution of 15,414 pictures for the training set, and 5138 for the verification and testing set. The training was carried out in 31 cycles. All images from the training set were used in each cycle.

The object detection results were satisfactory. The graphs in Figures 6–8 show the regression loss decreased significantly from the third cycle and classification loss stayed at a low level from the first cycle. The drops between values differed by the different dataset splits. It is necessary to note, that loss, regression loss, and classification loss are more significant for the dataset distributions 70/15/15 (Figure 7) and 60/20/20 (Figure 8). The loss function shows the disparity between the estimated value of the model iteration and its actual value. This disproportion is an estimate of the result of the measurement algorithm compared to its estimated result. In the case of RetinaNet, the Focal loss introduced in [24] is defined as the loss at the output of the classification subnetwork.

On the contrary, mAP (mean average precision) approached 1 (i.e., 100%) as the number of epochs increases, see Figure 6. The mAP varied more in the 70/15/15 (Figure 7), and 60/20/20 (Figure 8) distributions of the dataset. The overall results and values among epochs were slightly different but at the end of the learning process, they reached almost the same values as in the first defined dataset split.

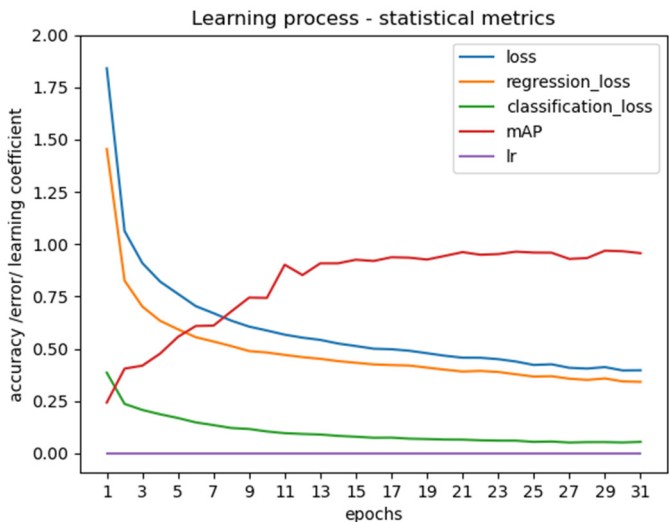

**Figure 6.** Results of learning process on a one-stage detector with 90/3/7 dataset split.

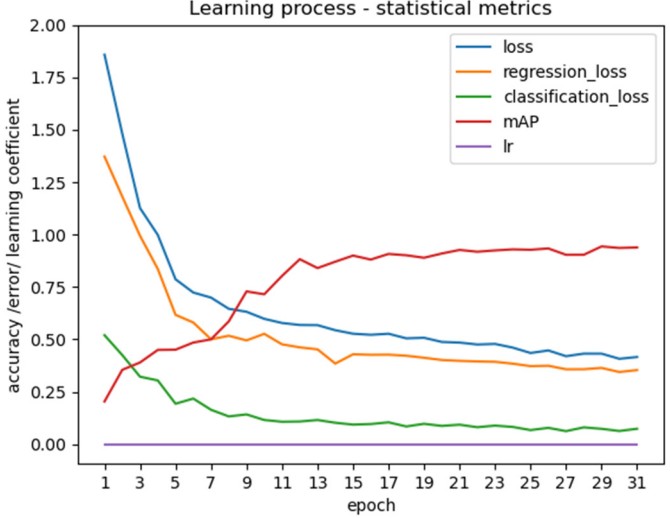

**Figure 7.** Results of learning process on a one-stage detector with 70/15/15 dataset split.

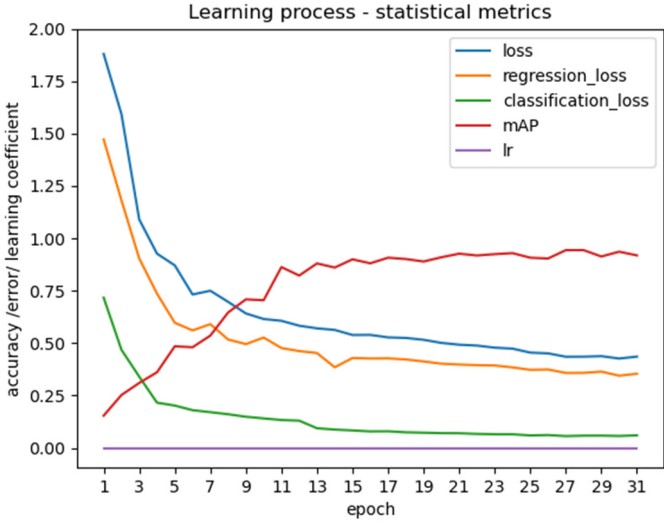

**Figure 8.** Results of learning process on a one-stage detector with 60/20/20 dataset split.

The graphs in Figures 9–11 show a more detailed view of the mAP results by vehicle category. The highest accuracy was achieved by the prediction of the vehicle category "CAR" with dataset split 90/3/7 in Figure 9. From cycle 9 onwards, it was above the 95% accuracy level. In order of accuracy, the "CAR" category was followed by the "VAN" and "BUS" categories. The last category in overall accuracy was the category called "TESLA", the reference vehicle of the TESLA brand, which belongs to the BEVs category. At the same time, for the "BUS" and "TESLA" categories, the mAP accuracies varied the most over the cycles.

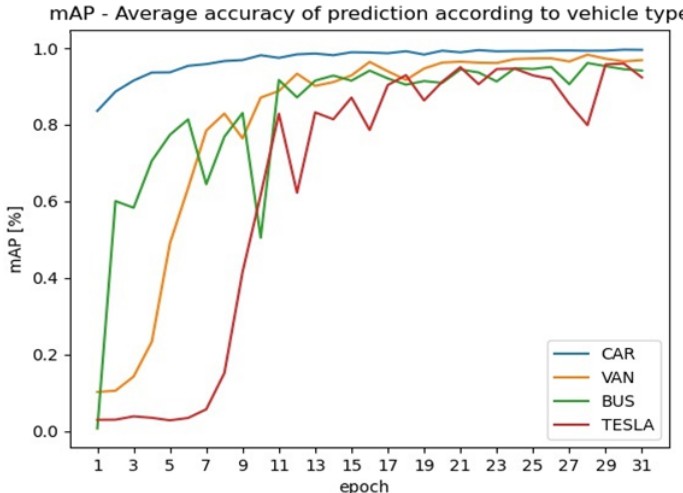

**Figure 9.** Accuracy of detection for each category with 90/3/7 dataset split.

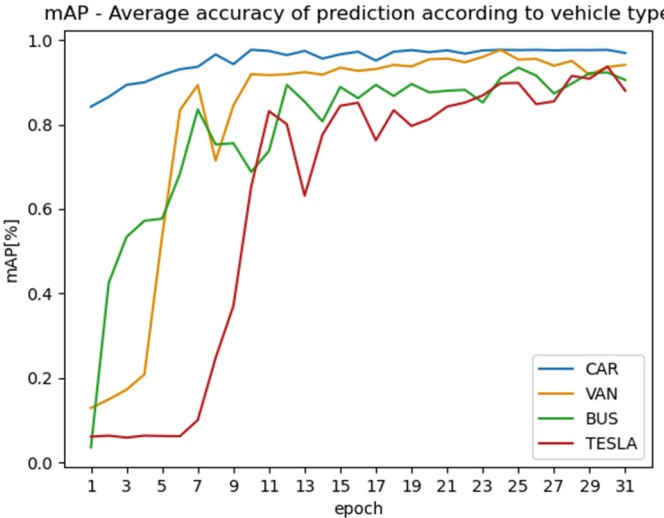

**Figure 10.** Accuracy of detection for each category with 70/15/15 dataset split.

The mAP results were slightly worse than in two other dataset splits (see Figures 10 and 11). The best mAP was achieved in the dataset split 70/15/15 for the category "CAR" as in the previous dataset distribution, followed by the categories "VAN", "BUS", and "TESLA". It was obvious that the values of mAP varied more during the cycles and their variation was only stabilized in the later epochs. The worse accuracy occurred for the category "TESLA" which had a very low mAP during the first 11 epochs and even afterwards. Large variations in the mAP values were also evident during cycles in the 60/20/20 dataset split. The main differences compared to the 70/15/15 dataset split occurred in the category "BUS" while in the "TESLA" category there were no evident changes.

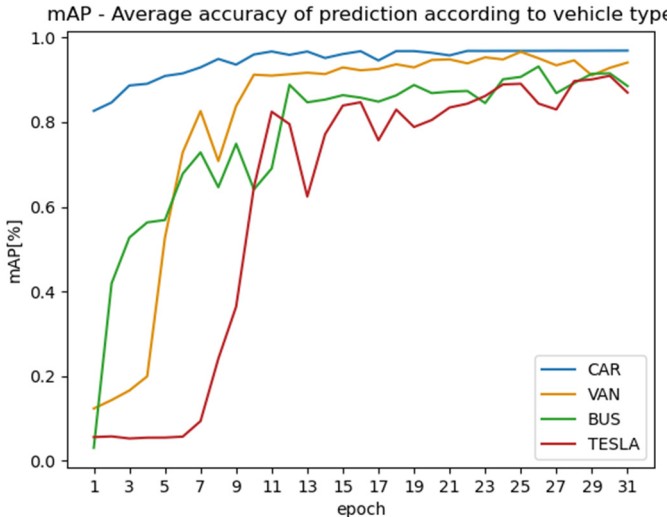

**Figure 11.** Accuracy of detection for each category with 60/20/20 dataset split.

The results for the dataset split 60/20/20 (see Figure 11) were worse than the first distribution. The order of the mAP values for the categories was the same as in the previous dataset splits. The mAP of the "CAR" category increased slowly and varied even more during epochs. The "BUS" category had low mAP until the 13th cycle when it reached 80%. The less affected categories among the dataset splits were "CAR" and "VAN". On other hand, it was obvious that the worst accuracy occurs for the categories "BUS" and "TESLA".

The trained object detector was used in real-time detection as a part of a complex traffic lights control system. A comparison between the original image and the image with the detected and classified vehicles can be seen in Figure 12. The object detector distinguished ICEVs and BEVs from the thermal stream recorded by the camera mounted on the mast arm of the traffic light. The system accordingly selected the ratio of detected BEV and ICEV and a pre-prepared signal plan for the optimization of traffic flow through the arterial traffic light of a suburban area. The testing was performed on the simulation traffic light controller while the thermal video was streamed from the real traffic light intersection location and sending to the corresponding server.

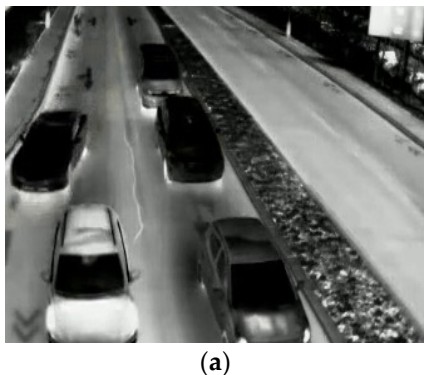
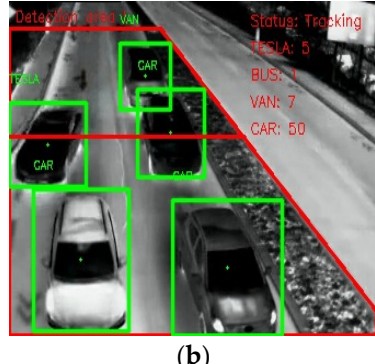

(**a**)                                                                 (**b**)

**Figure 12.** Comparison frames without detection (**a**) and with detection (**b**).

Statistical evaluation was performed to assess the difference in the number of detection windows in the image for different settings of the algorithm threshold parameters mentioned in the previous section. First, it was necessary to verify the normality of the data using the Shapiro–Wilk test and the Kolmogorov–Smirnov test. The graphical representation and the calculated values do not correspond to a normal distribution (see Figure 13). The methods of nonparametric tests need to be adapted by the Kruskal–Wallis test and the median test [67]. These tests are a nonparametric variant of the analysis of variance.

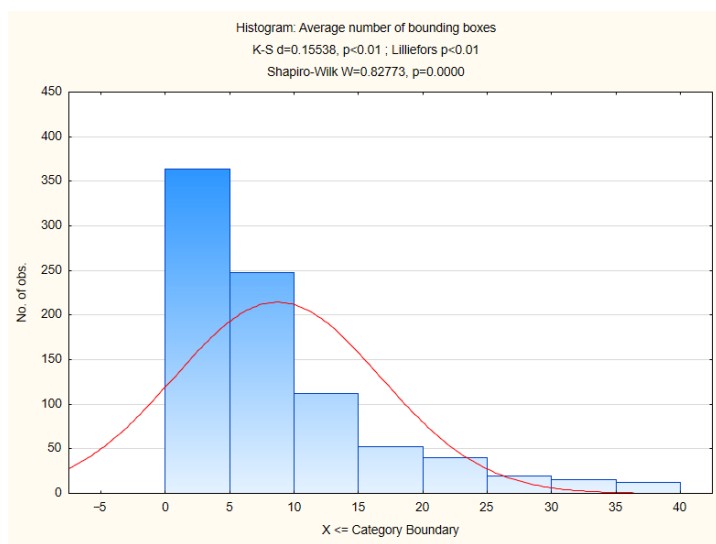

**Figure 13.** Test of normality.

The statistical test was based on the rejection of the null hypothesis, which assumes equal median numbers of detected temperature fields (detected by MSER) for both vehicle categories. To verify this, it was necessary to compare the calculated statistical value of the H test with the threshold value, which was determined by the chi-square.

The comparison of *H* for the Kruskal–Wallis test, *H* = 102.2524 and chi-square = 42.66667, subsequently gives:

$$H > \chi^2 \tag{8}$$

We can now reject the null hypothesis and declare that at the level of asymptotic significance *p* = 0.05, there is a significant difference between the average number of detected windows in the temperature fields of the vehicle categories of interest. The box plot below (Figure 14) represents this difference in the average number of detection windows between ICEVs and BEVs in graphical form. The average number of detection windows for BEVs was noticeably lower than for ICEVs. The graph includes the calculated upper and lower 95% confidence intervals, where SE represents the standard error for the sample mean, and the value of 1.96 is the approximate value of the 97.5 percentile point normal distribution.

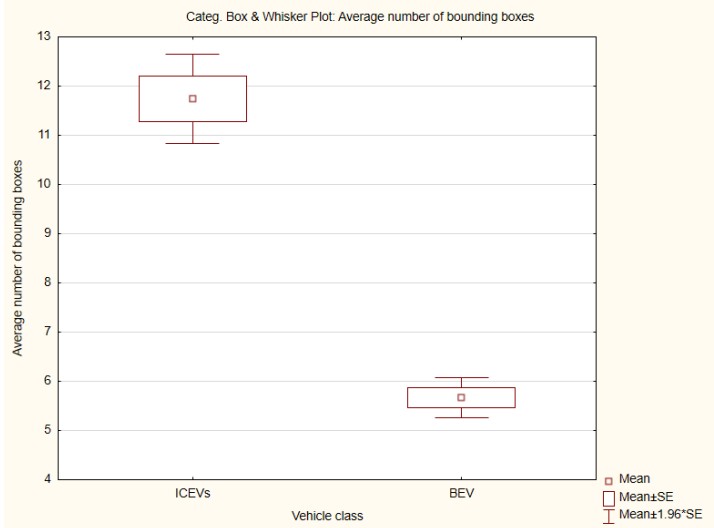

**Figure 14.** Box plot of average number of bounding boxes.

Figure 15 shows an example of the detection of ICEV and BEV thermal fields. Figure 15a illustrates the detected heated combustor mask, as well as other detected MSERs. In a portion of Figure 15b, the detected BEV MSERs can be seen, but their ratio is lower compared to ICEVs. The difference is attached here as follows, as an example of the result after applying the MSER algorithm. By using the tools from the OpenCV library that have been added to the object detector and thus cropping the images to the shape of the vehicle only, the disturbing elements located around the vehicle that could be detected by the MSER algorithm when applied to the whole image were limited. It may not only be additional heat sources that offset the boundaries of the mixed pixels that are detected. It could also be interference caused by low image resolution, which due to noise and distortion, creates mixed pixels in objects and parts of the image farther away from the camera location. Subsequently, these areas are detected as MSERs using the same thresholding parameters.

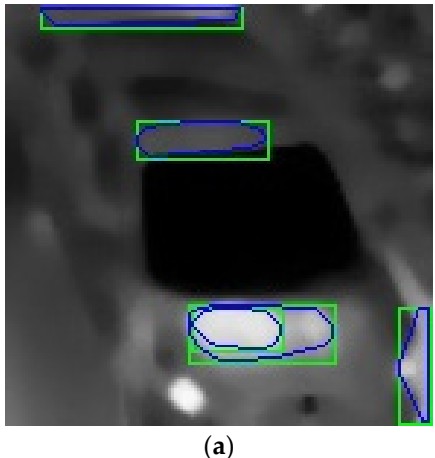 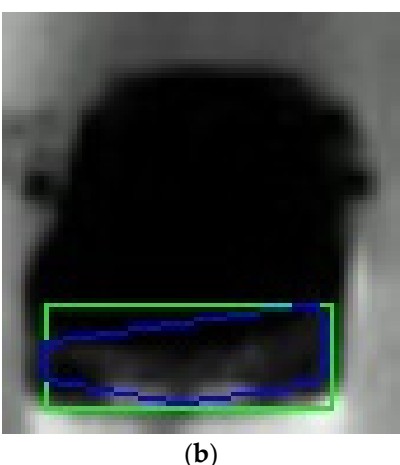

(**a**)　　　　　　　　　　　　　　　　(**b**)

**Figure 15.** Example of results of MSER detection: (**a**) MSERs detection of ICEV; (**b**) MSERs detection of BEV.

## 5. Discussion

The combination of the RetinaNet object detector and the MSER algorithm to detect thermal fields seems to be a viable solution for the classification of vehicle propulsion systems. The reason for the declared database splits was to test the hypothesis of whether it is possible to distinguish between BEV and ICEV in a thermal video stream under real traffic conditions using object detection methods. The object detector learns using a loss function. If the deviation between the actual values and the expected values is too large, the loss function shows high values. This can lead to huge prediction errors. However, by using optimization functions, the loss function is trained to reduce prediction errors. The regression loss represents the results for the root mean square error between the estimated value and the actual value. On the contrary, classification loss shows how the classification model performs when the probability value is between 0 and 1. It shows the variation between two probability distributions and the relative entropy between them [68]. The defined distributions of the dataset met the requirements of the model and corresponds to its needs for the verification of the above-defined hypothesis. The reasons we chose to mount the thermal camera on the column arm of the traffic lights instead of on the tunnel portal are mainly due to the difficulties in placing equipment (cameras) that must be connected to the Ethernet network and the power supply, and due to cybersecurity [69], the lack of cable reserves, and untested technology in the tunnels including obtaining the necessary permits from the authorities. According to the design documentation and legislation in the Czech Republic for tunnels, the equipment in tunnels is classified as critical infrastructure elements and therefore of high security, including fire protection and so on. However, we have taken the liberty to use the given percentage distribution of images into individual database sets, not only because of the used literature [23], but also because of the expected application of the technology in tunnels or in tunnel complexes. We know

that with fewer test/validation data, our model evaluation/statistical indicators of model performance have greater variance. The tunnel provides a much more stable environment with still conditions of illumination, temperature, humidity, and stability of the monitored road environment. For this reason, we used fewer images in our solution for the validation and test set, as higher quality thermal recordings in tunnels are expected and hence higher detection accuracy due to easier recognition of image features. In the future, we anticipate further research that will take into account technologies for outdoor conditions. Another limitation might be that the object detector was trained only for the TESLA model S as a representative of BEV category. For a completive operation of the system, it is necessary to test it on more BEV models, in different environments, and changing number of passengers.

The constructed database is then used as the basic input for the MSER algorithm that solves the task of distinguishing vehicle drives. RetinaNet's computation time was close to real-world imaging values, and the MSER algorithm outperformed these values. Therefore, its interconnection is appropriately set up. A system with such detection accuracy and information complexity, i.e., with a defined vehicle type including the used propulsion, can subsequently be applied to a variety of real-world applications. The application seems to be viable as a diagnostic, preventive, and warning system for tunnels. Such a system might predictively assess vehicles entering tunnel tube spaces. Its main purpose is to analyse vehicles according to the range of detected temperature fields of the vehicle masks and to prevent entry and prevent tunnel fires in time. Fire prevention is particularly relevant for BEVs which are very difficult to extinguish with commonly available extinguishing methods. Another preventive measure is detecting the thermal side profile of vehicles. It might warn of trucks with an overheated brake system and prevent traffic accidents with catastrophic scenarios in tunnels. The passenger detection and total occupancy of vehicles should also be taken into account in the case of fires in tunnels.

The proposed detection can be achieved by detecting large MSERs, i.e., the temperature field in BEVs. In the case of BEVs, such a condition would represent an overheating vehicle. Indeed, the results and data from this and previous research show that the number of BEV MSERs decreases when the value of the minimum area parameter increases. In other words, the number of detection windows is lower for BEVs with the same thresholding parameters of the MSER algorithm compared to ICEVs. The disadvantage of the MSERs detection method is a higher percentage of misleading heat sources around the vehicle which might lead to false detections. The algorithm evaluates these areas as areas with similar pixel information using the same thresholding algorithm parameters. When another additional classifier is used, false positives can be discarded to ensure adequate classification based on the number of detection windows with their predefined threshold size for each category.

Additionally, this system can be used in the case of fire in smoky tunnel areas to identify the burning vehicle. Future research will focus on the detection of hybrid vehicles. The current object detector detects hybrid vehicles as an ICEV. A wider dataset of hybrid vehicles is necessary to tune the parameters for their detection by the MSER algorithm. The possibility of using thermal cameras for side or rear detection of the vehicle requires more accurate data. Detecting and providing information on the number of passengers in the vehicle in the tunnel is important for emergency services in case of an emergency in the tunnel (accident, fire). This method of detection might significantly develop new ITS trends in tunnels, especially adding data for cooperative C-ITS systems [11]. Last, but not least, the security of data transmission in the tunnel system, their protection in case of detection of persons, and ensuring cyber security are burning issues which need to be investigated [69].

Other applications are evident in vehicle preference systems in urban agglomerations for controlling traffic lights, possibly using them for preference of BEVs, in the context of area control and entry restrictions for combustion vehicles, and for toll systems based on the distinction between BEVs and ICEVs. The detection of idling cars in traffic jams is another possible direction for further research. Table 2 below describes the rest of the key findings of the proposed method against the state-of-the-art methods.

**Table 2.** Key findings of the proposed method.

| Key Findings | Proposed System |
|---|---|
| Vehicle detection | Classification of the type (BEV/ICEV) of vehicle |
| Data verification | Statistical confirmation of thermal differentiation of BEV and ICEV propulsions |
| Innovation | Concept of thermal detection and usage of MSER alg. for classification of vehicle propulsions |
| Applicability | Testing and operation in real traffic conditions |

The next phases include:

- Complementing the MSER algorithm with an MSER region classifier using methods such as SVM, Hu moments and Random Naïve Bayes (RNB), etc.
- Extending the database with other BEV manufacturers and include other vehicle types such as trucks and deepening the quality of the object detector training and its accuracy results.
- Acquisition of infrared imagery and thermal data from tunnels and validation of the assumptions and hypotheses of use for security detection.
- Validation of vehicle detection in the categories of trucks and goods transport can have a major positive effect on safety in critical infrastructure, especially in tunnel structures, not only in terms of vehicle failures, semi-trailers, etc. but also in terms of goods transported. For example, the transport of dangerous goods under the ADR (Accord Dangereuses Route) agreement.

## 6. Conclusions

The results show that the method of combining RetinaNet and the MSER algorithm is a suitable system for vehicle detection and classification from a thermal stream. Its advantage is the possibility of use in real traffic systems due to its fast computation time. The linking is effective for obtaining comprehensive information on the detected vehicle, i.e., the vehicle category as BEV or ICEV. The higher accuracy of the object detector represents a system that can operate in constant conditions e.g., in tunnels. The results of the object detector with the highest achieved mAP of 96.9% showed that the built architecture can recognize vehicles and easily categorize them. The extension of the system to classify the vehicles used thermal point detection to distinguish vehicles into BEVs and ICEVs. The MSER algorithm was used to implement it. This method has reached its testing and refinement phase. However, the statistic results proved the existence of the significant difference between the number of MSERs of the BEV and the ICEV categories. The issue of categorization of vehicle propulsions based on the confirmed hypothesis needs to be further elaborated. The researchers are focusing on removing its weaknesses and extending its deployment in ITS applications.

**Author Contributions:** Data curation, D.Š.; methodology, D.Š., T.T. and M.R.; project administration, T.T.; formal analysis, D.Š. and T.T.; resources, T.T., D.Š. and M.R.; writing—original draft preparation, D.Š. and T.T.; writing—review and editing, D.Š., T.T., M.R. and P.I. All authors have read and agreed to the published version of the manuscript.

**Funding:** This research was funded by the Faculty of Transportation Sciences and Department of Vehicles and Ground Transport, CULS Prague, and the Faculty of Engineering, No. IGA TF 2021:31150/1312/3113.

**Institutional Review Board Statement:** Not applicable.

**Informed Consent Statement:** Informed consent was obtained from all subjects involved in the study.

**Data Availability Statement:** Restrictions apply to the availability of these data. Data were obtained from the company Eltodo and it is available from the authors after an official request. These data would be available only with the permission of the third party—company Eltodo from the Czech Republic.

**Acknowledgments:** This research was supported by the Department of Transport Telematics CTU in Prague, Faculty of Transportation Sciences and Department of Vehicles and Ground Transport, CULS Prague, and the Faculty of Engineering.

**Conflicts of Interest:** The authors declare no conflict of interest. The funders had no role in the design of the study; in the collection, analyses, or interpretation of the data; in the writing of the manuscript, or in the decision to publish the results.

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
