# Peer review of "Use of One-Stage Detector and Feature Detector in Infrared Video on Transport Infrastructure and Tunnels"

_sustainability, doi:10.3390/su15032122_

Round 1

Reviewer 1 Report (Previous Reviewer 3)

The authors have answered all my questions, and the points raised for reviewer 1 were very interesting and partially answered by the authors. The low rate of BEV identified by the algorithm is something to worry about, but this is stated in the limitations.  Anyway, I think the manuscript is much better now.

Some minor review:

-page 2, line 79. Please change all "chapter" words to "section" (check this in the entire manuscript - eg:pag7, line 266). Section 2 is missing in this explanation. 

-page 7, line 240. The correct phrase is: "In this paper, a scientific problem was stated as follows: “Is it possible to use the method of object detection for categorization of BEVs and ICEVs based on the thermal fields.”".

Author Response

-page 2, line 79. Please change all "chapter" words to "section" (check this in the entire manuscript - eg:pag7, line 266). Section 2 is missing in this explanation.

We changed all the words “chapter“ to “section”. Section 2 is described in the first sentence of the revised paragraph: “The following section will discuss the approaches of object detection using convolutional neural networks and object resolution based on their heat signatures.”

-page 7, line 240. The correct phrase is: "In this paper, a scientific problem was stated as follows: “Is it possible to use the method of object detection for categorization of BEVs and ICEVs based on the thermal fields.”".

We corrected this sentence.

Reviewer 2 Report (Previous Reviewer 2)

The authors have modified the article as recommended, but the figures 3, 5, 12 and 15, and the table 1 still extends beyond the page boundaries.

Author Response

The authors have modified the article as recommended, but the figures 3, 5, 12 and 15, and the table 1 still extends beyond the page boundaries.

We have followed the Microsoft Word template and the styles contained in it. According to this template, we are within the page boundaries.

Reviewer 3 Report (New Reviewer)

The work presented by the authors focuses on using the one-stage detector and feature detector in an infrared video on transport infrastructure and tunnels. The manuscript is well-written and easy to follow, and the authors have well-thought-out main contributions. The theoretical background is concrete, complete, and correct, and the authors have provided all the intermediate steps to enable the average reader to follow them easily. However, minor revisions are requested to improve the quality of the manuscript.

1. The key contributions of the paper should be presented in bullet form for easy tracking.

2. The milestones in object detection in Figure 1 need to be extended to 2022. 

3.  A Table summarizing the key findings compared to related state-of-the-art needs to be included. 

4. The implications of the projected results and their usefulness need to be added to the paper.

5. What are the limitations of the current study? 

6. The future scope should be added to the conclusion.

Author Response

This manuscript is a resubmission of an earlier submission. The following is a list of the peer review reports and author responses from that submission.

Round 1

Reviewer 1 Report

  1. English needs revision
  2. The abstract needs to be improved; the problem actuality is not defined. The advantage of the proposed method compared to state of art is not provided.
  3. The introduction should be rewritten. Right now, it starts with a description of the proposed system without problem formulation.
  4. Authors use ITS without a description of this term.
  5. The authors state, „ The widely used visual cameras still have low accuracy in bad weather conditions such as fog, snow or heavy rain. Therefore, it is necessary to replace or supplement 43 conventional methods using visual cameras with infrared cameras“. In practice, augmentation could solve this issue. The main problem would be “physical“ lens coverage with dust or snow, and the proposed method is also camera-based.
  6. The literature review proposed in the introduction section does not provide state-of-the-art.
  7. Well-known information should be deleted from a paper, for example: „Vehicles use coolant-filled radiators for engine cooling needs. The radiators heat up as the coolant passes through and their temperature is partially proportional to the engine load and temperature. This characteristic can be exploited to focus the detection on the radiator grille. Due to the running of the combustion engine, both the engine itself and the other auxiliary parts of the engine are heated, and this heat is emitted to neighbouring parts such as the mask, hood and side of the vehicle.“
  8. „The authors in [31] state that the electric motor and the components required for its operation are heated only to a very low level compared to the internal combustion engine. This difference could be used as a basis for detecting BEVs.“ What if the ICE vehicle just started and did not reach working temperature?
  9. Figure 2 should be described the information provided in figures a) and b) is needed
  10. „Vehicle classification using CNN, which was trained using a database of 262 images acquired by a high-resolution visual camera.“ Why is this needed if you use a different camera type and low resolution?
  11. „The vehicle chosen to represent BEVs was a Tesla Model S and the category was named TESLA.“
  12. Currently, there are more than 40 electric vehicle manufacturers; why only Tesla is included? What about hybrid vehicles? How are they classified?
  13. Figure 9 is not clear
  14. There is no text on page 12
  15. In the section Analysis and results, the dataset was not prepared properly; authors used 23,258 images for training (more than 90%), 7% for a test, and about 3% for verification. First of all, it should be training, validation, and testing. The second point is that such a distribution will lead to overtraining. My proposal would be to start from 70%/15%/15% and repeat the test changing this proportion to 60%/20%/20%. Also, it is not clear how the dataset was created.
  16. Conclusions are not accurate, you can‘t declare that „The more than 95% accuracy of the object detector represents a robust system that 575 can be deployed in live operation after further testing in different environments.“ Other manufacturers of electric vehicles were not taken into account; hybrid vehicles were not taken into account, a problem when vehicles with ICE which just started their journey were not pointed. 
  17. The scientific novelty is not clear, description of the dataset brings a lot of doubts regarding its accuracy. Authors used NN developed by other authors, or was it modified? Such a system needs to perform in real-time; however, these aspects were not addressed in the paper.

Reviewer 2 Report

Some Recommendations to the paper:

 1.      The organization of the paper should be given in the last paragraph of Introduction section. The objective and originality of the paper should be given in the introduction section.

2. The manuscript needs to be checked for spelling errors in the English language such as “algorithmicizing”.

3. The formulas (4) and (5) are identical.

4. It is necessary to add a reference to ‘’From the graphical representation and also from the calculated values, which are part of the histogram in Figure 15, it follows that the data do not come from a normal distribution, and therefore it is necessary to adapt the methods of nonparametric tests, namely the Kruskal-Wallis test and the median test. “

5. More suggestions for future research should be added to the Conclusion section. Limitations of the work should be mentioned in the conclusion section.

Reviewer 3 Report

The manuscript shows an interesting analysis of vehicles detector. The literature review is quite complete, the analysis is robust and the conclusions are connected to the analysis.

I just have a minor review:

-pag4, line 131. The reference "Dong" can be deleted, because the reference [22] is already there.

-pag4, line 134. The reference "Bautista et al." can be deleted, because the reference [23] is already there. 

-pag4, line 138. The reference "Fan et al." can be deleted, because the reference [24] is already there. 

-pag4, line 141. The reference "Chu et al." can be deleted, because the reference [25] is already there. 

-pag4, line 148. The reference "Li et al." can be deleted, because the reference [27] is already there. 

-pag5, line 185. The reference "Kuchár et al." can be deleted, because the reference [35] is already there. 

-pag5, line 193. The reference "Nister and Stewénius" can be deleted, because the reference [37] is already there. 

-pag10, line 355. The reference "Matas et al." can be deleted, because the reference [36] is already there. 

Round 2

Reviewer 1 Report

My opinion is the same regarding the paper. The work is very weak.  The authors made only text corrections without improving the methodology.   Please see my previous comment: 15. In the section Analysis and results, the dataset was not prepared properly; authors used 23,258 images for training (more than 90%), 7% for a test, and about 3% for verification. First of all, it should be training, validation, and testing. The second point is that such a distribution will lead to overtraining. ... Authors answer: We thank you for your advice. The work and results presented in the article were part of pilot study. We will apply the proposal split of the dataset in further research.    Conclusion: current results are not reliable.    From my point of view, the rest comments were not addressed properly.
